# Effective Littlestone dimension

## Abstract

Delle Rose et al. (COLT'23) introduced an effective version of the Vapnik-Chervonenkis dimension, and showed that it characterizes improper PAC learning with total computable learners. In this paper, we introduce and study a similar effectivization of the notion of Littlestone dimension. Finite effective Littlestone dimension is a necessary condition for computable online learning but is not a sufficient one—which we already establish for classes of the effective Littlestone dimension 2. However, the effective Littlestone dimension equals the optimal mistake bound for computable learners in two special cases: a) for classes of Littlestone dimension 1 and b) when the learner receives as additional information a bound on the numbers to be guessed. Interestingly, finite effective Littlestone dimension also guarantees that the class consists only of computable functions.

**Keywords:** Online learning, Littlestone dimension, computable machine learning

## 1. Introduction

Two fundamental models of machine learning, PAC learning and online learning, have been recently revisited from the viewpoint of computability theory (Agarwal et al. (2020), Sterkenburg (2022), Delle Rose et al. (2023), Hasrati and Ben-David (2023)). In the classical setting, a learning algorithm is understood as a function, getting a sample $S$ and an input $x$ and outputting its prediction of the value on $x$. Although this is called an "algorithm", it is not assumed to have a Turing machine that computes it. The existence of a learning algorithm for a hypothesis class can then be characterized by a combinatorial dimension of that class, namely, the VC dimension in the case of PAC learning and the Littlestone dimension in the case of online learning.

What if we do require a learning algorithm to be computable by a Turing machine? We obtain "computable counterparts" of PAC and online learning models that might no longer be characterized just by a combinatorial dimension. For instance, Sterkenburg (2022) constructs a class with finite VC dimension, given by a decidable set of functions with finite support, that has no computable PAC learner, even if the learner is allowed to be improper (output functions outside the class). Likewise, Hasrati and Ben-David (2023) observe that there is a class that has Littlestone dimension 1 and consists of finitely supported functions but does not have an online learner, computable by a partial Turing machine (it might not halt on non-realizable inputs) with finite number of mistakes.

Maybe for a characterization of a computable version of PAC or online learning, it is enough to "effectivize" the corresponding combinatorial notion? One instance when the answer is yes has been established by Delle Rose et al. (2023) for computable PAC learning. To this end, they introduced the notion of the effective VC dimension of a hypothesis class $H$. The usual VC dimension is defined as the maximal size of a subset of the domain where functions from $H$ can realize all dichotomies. Dually, one can define this notion as the minimal $d$ such that for any subset of size $d + 1$ there exists a dichotomy, not realizable by $H$. In the effective version of VC dimension, there must be a Turing machine that, given an $(d + 1)$-size subset, outputs a dichotomy, not realizable by $H$. The minimal $d$ for which such a Turing machine exists is called the effective VC dimension

of $H$. As Delle Rose et al. (2023) show, classes admitting a computable PAC learner are exactly classes having finite effective VC dimension. An important detail here is that they assume that the learner has to be computed by a *total* Turing machine (another possibility is when a Turing machine might not halt on some samples that are not realizable by the class).

In this paper, we introduce a similar "effectivization" of the Littlestone dimension and study its relationship with the computable online learning. The usual Littlestone dimension of a hypothesis class $H$ is defined as the maximal $d$ for which there exists a depth-$d$ Littlestone tree with every branch realizable by $H$. Following the idea of Delle Rose et al. (2023), we define the effective Littlestone dimension of $H$ as the *minimal* $d$ for which there exists a Turing machine that, given a Littlestone tree of depth $d + 1$, indicates a branch not realizable by $H$.

Our contribution with respect to the effective Littlestone dimension consists of the following.

- In a similar manner, we define the notion of the effective threshold dimension and observe that classes with finite effective Littlestone dimension coincide with classes of finite effective threshold dimension.

- We observe that a class that admits an online learner, computable by a total Turing machine (for brevity, a total computable online learner), that makes at most $d$ mistakes, has effective Littlestone dimension at most $d$.

- We show that the converse does not hold. We construct a class of effective Littlestone dimension 2 that does not admit even a partial computable online learner ("partial" meaning that the Turing machine, computing it, might not halt on some non-realizable samples) with a finite number of mistakes.

- On the positive side, we consider a weaker game, in which the Adversary can only give numbers bounded by some constant and show that effective Littlestone dimension characterizes computable learnability in this setting.

- We also show that every class of finite effective Littlestone dimension consists of computable functions. As a consequence, every class of effective Littlestone dimension 1 admits a total computable online learner with 1 mistake.

Similar failure of the combinatorial characterization of computable learning was recently observed by Gourdeau et al. (2024) for computable robust PAC learning.

## 2. Preliminaries

By *hypothesis classes* we mean sets of functions from $\mathbb{N}$ to $\{0, 1\}$. By *samples* we mean finite sequences of pairs from $\mathbb{N} \times \{0, 1\}$. A sample $S = (x_1, y_1) \ldots (x_k, y_k)$ is *consistent* with a function $f : \mathbb{N} \to \{0, 1\}$ if $f(x_1) = y_1, \ldots, f(x_k) = y_k$. A sample $S = (x_1, y_1) \ldots (x_k, y_k)$ is *realizable* by a hypothesis class $H$ if there is a function in $H$ with which $S$ is consistent.

A *learner* is a partial function $L : (\mathbb{N} \times \{0, 1\})^* \times \mathbb{N} \to \{0, 1\}$ (thus, the first input to $L$ is a sample and the second input is a natural number). We say that a learner $L$ is a learner for a hypothesis class $H$ if $L(S, x)$ is defined for every sample $S$, realizable by $H$, and for every $x \in \mathbb{N}$. A total learner is a learner which is defined everywhere. By default, a learner can be partial, but sometimes, to stress that a statement applies not only to total learners, we write "partial learner".

A learner $L$ is computable if there exists a Turing machine that outputs $L(S, x)$ on $(S, x)$ on which $L$ is defined, and does not halt for $(S, x)$ on which $L$ is not defined

For a given sample $S$, the learner induces a (possibly, partial) function $L_S \colon \mathbb{N} \to \{0, 1\}$ by setting $L_S(x) = L(S, x)$, to which we refer as the hypothesis of $L$ after the sample $S$.

A learner $L$ for a hypothesis class $H$ is called an *online learner* for $H$ *with at most $d$ mistakes* if for any $H$-realizable sample $S = (x_1, y_1) \ldots (x_k, y_k)$ there exists at most $d$ of $i \in \{1, \ldots, k-1\}$ such that

$$L((x_1, y_1) \ldots (x_i, y_i), x_{i+1}) \neq y_{i+1}.$$

**Lemma 1** *Let $H$ be a hypothesis class an $d$ $L$ be an online learner for $H$ with at most $d$ mistakes, for some $d \in \mathbb{N}$. Then every function $f \in H$ coincides with $L_S$ on some sample $S$, consistent with $f$.*

**Proof** Indeed, if there is no such sample, we can construct a sample, consistent with $f$, on which $L$ makes more than $d$ mistakes. Namely, we start with the hypothesis of $L$ after the empty sample. It disagrees with $f$ on some $x_1 \in \mathbb{N}$ which we put to the sample as $(x_1, f(x_1))$, causing the first mistake. The hypothesis of $L$ after $(x_1, f(x_1))$ disagrees with $f$ on some $x_2$, and we add this $(x_2, f(x_2))$ to the sample, forcing the second mistake. In this way, we can force arbitrarily many mistakes. ■

This lemma implies that classes of finite Littlestone dimension are always countable, and that classes that admit a computable online learner (even when it is not total) consist only of computable functions.

By a *Littlestone tree* of depth $d$ we mean a complete rooted binary tree of depth $d$ where: (a) edges are directed from parents to children, with each edge labeled by 0 or 1 such that every non-leaf node has one out-going 0-edge and one out-going 1-edge; and (b) non-leaf nodes are labeled by natural numbers. Every edge in such a tree can be assigned a pair $(x, y) \in \mathbb{N} \times \{0, 1\}$ where $x$ is the label of the node this edge start at and $y$ is the bit, labelling this edge. Thus, every directed path in this tree can be assigned a sample, obtained by concatenating pairs, assigned to its edges.

The *Littlestone dimension* of a class $H$, denoted by $\mathsf{Ldim}(H)$, is the minimal $d \geq 0$ such that in every $(d + 1)$-depth Littlestone tree $T$ there exists a leaf such that the sample, written on the path from the root to this leaf, is not realizable by $H$. The *effective Littlestone dimension* of a class $H$, denoted by $\mathsf{effLdim}(H)$, is the minimal $d \geq 0$ for which there exists a total Turing machine that, given as input a complete rooted binary tree of depth $d + 1$, outputs a leaf of this tree such that the sample, written on the path to this leaf from the root, is not realizable by $H$.

**Proposition 2 ((Littlestone, 1988))** *For any class $H$, the minimal $d \geq 0$ for which there exists an online learner for $H$ with at most $d$ mistakes is equal to $\mathsf{Ldim}(H)$.*

If $H$ is a hypothesis class, then for $x \in \mathbb{N}$ and $b \in \{0, 1\}$, by $H_b^x$ we denote the class $\{f \in H \mid f(x) = b\}$.

**Proposition 3** *(Littlestone, 1988) For any hypothesis class $H$ of finite positive Littlestone dimension, and for every $x \in \mathbb{N}$, either $H_0^x$ or $H_1^x$ have smaller Littlestone dimension than $H$.*

We say that a learner $L$ for a class $H$ *PEC-learns* it if for every $f \in H$ and for every probability distribution $D$ over $\mathbb{N}$ the following holds. If we consider an infinite sequence

$$(X_1, f(X_1))(X_2, f(X_2))(X_3, f(X_3)) \dots$$

where $X_1, X_2, X_3 \dots$ are sampled independently from $D$, then with probability 1 there exists $n_0$ such that for all $n \geq n_0$, the hypothesis of $L$ after the sample $(X_1, f(X_1)) \dots (X_n, f(X_n))$ coincides with $f$ on all $x \in X$ that have positive probability w.r.t. $D$.

Given a learner $L$ and a sample $S = (x_1, y_1) \dots (x_k, y_k)$, the number of *mind changes* of $L$ on $S$ is the number of $i \in \{0, 1, \dots, k-1\}$ such that the hypothesis of $L$ after $(x_1, y_1) \dots (x_i, y_i)$ differs from those after $(x_1, y_1) \dots (x_{i+1}, y_{i+1})$. We say that a learner $L$ for a class $H$ makes at most $d$ mind changes on it if the number of mind changes of $L$ on every $L$-realizable sample does not exceed $d$.

## 3. Effective threshold dimension

Shelah (1982) found a deep connection between the Littlestone dimension of a class and the number of *thresholds* which are contained by such class. Here we use a reformulation due to Alon et al. (2022).

Let $k \in \mathbb{N}$. We say that a hypothesis class $H$ *contains $k$ thresholds* if there are $x_1, \dots, x_{k-1} \in \mathbb{N}$ and $h_1, \dots, h_k \in H$ such that

$$\forall i, j \leq k, \quad h_i(x_j) = 0 \Leftrightarrow i \leq j.$$

We then say that $H$ has *threshold dimension $t$* if $t$ is the largest number for which $H$ contains $t$ thresholds.

**Theorem 4 (Shelah (1982); Alon et al. (2022))** *For any hypothesis class $H$ and $d \geq 0$:*

1. *if $\mathsf{Ldim}(H) \geq 2^d$, then $H$ has threshold dimension at least $d$;*

2. *if $H$ has threshold dimension at least $2^d$, then $\mathsf{Ldim}(H) \geq d$.*

*Thus, $H$ is online learnable if and only if $H$ has finite threshold dimension.*

We now establish the effective counterpart of the previous result. Let us define the *effective threshold dimension* of a hypothesis class $H$ as the minimal $d \geq 0$ for which there is a total Turing machine $w$ which, on input any $d+1$ points $x_0 \leq \cdots \leq x_d \in \mathbb{N}$, outputs a threshold (namely a binary word of the form $0^i 1^{d-i}$ for $i \in \{0, \dots, d\}$) $\tau$ such that $h(x_0) \dots h(x_d) \neq w(x_0, \dots, x_d)$.

**Theorem 5** *Let $H$ be any hypothesis class. Then $H$ has finite effective Littlestone dimension if and only if it has finite effective threshold dimension.*

**Proof** Assume that $H$ has effective Littlestone dimension smaller than $d$, for $d \geq 1$, as witnessed by the algorithm $w$. We now show that we can turn $w$ into an algorithm witnessing that the threshold dimension of $H$ is at most $2^d - 1$. On input $x_0, \dots, x_{2^d-2} \in \mathbb{N}$, we construct a Littlestone tree $T$ of depth $d$ corresponding to all thresholds on $x_0, \dots, x_{2^d-2}$ as follows. $T$, in fact, corresponds to the binary search tree over $0, \dots, 2^d - 2$: the root of $T$ gets label $x_{2^{d-1}-1}$ and, if a node at distance

$0 \le n < d$ has label $x_m$, then its left child gets label $x_{m+2^{d-m-1}}$, while its right child gets label $x_{m-2^{d-m-1}}$. Clearly, the $i$-th leftmost root-to-leaf path in $T$ corresponds to the threshold $0^i 1^{d-1}$. Therefore, it suffices to compute $w(T)$ and output the threshold corresponding to the path indicated by $w(T)$.

Now assume that $H$ has effective threshold dimension smaller than $t$, as witnessed by the algorithm $w$. By Theorem 4, $\mathsf{Ldim}(H) < 2^t$. We claim that the effective Littlestone dimension of $H$ is also smaller than $2^t$. Given a Littlestone tree of depth $2^t$, let $S$ be the set of all labels of nodes in $T$. We compute the finite class

$$H' = \{f \in \{0,1\}^S \colon (\forall x_0, \dots, x_t \in S)\, f(x_0) \dots f(x_t) \neq w(x_0, \dots x_t)\},$$

namely the class of Boolean functions defined on $S$ which respect the constraints imposed by $w$. Observe that $w$ witnesses that also the effective threshold dimension of $H'$ is smaller than $t$: hence, by Theorem 4, its Littlestone dimension is less than $2^t$. Thus, $T$ must contain a root-to-leaf path whose corresponding sample is not realized by $H'$, and we can find it effectively by simply checking all the finitely many functions from $H'$. But such sample cannot be realized by $H$ either, as $H'$ contains all functions in $\{0,1\}^S$ which are restrictions of hypothesis from $H$ to the set $S$, since every hypothesis in $H$ agrees with $w$. ∎

## 4. Effective Littlestone dimension vs. computable online learning

**Proposition 6** *For any hypothesis class $H$ and for any $d$, we have the following. If $H$ admits a total computable online learner which makes at most $d$ mistakes, then the effective Littlestone dimension of $H$ is at most $d$.*

**Proof** Let $L$ be a total computable online learner for $H$ with at most $d$ mistakes. Given a $(d+1)$-depth Littlestone tree $T$, we find a leaf of it on which $L$ makes $d+1$ mistakes. Namely, we give $L$ the number from the root, wait for its predictions (by totality of $L$, we will always receive it), go to the child which contradicts this prediction, give the number from this child, and so on. The sample on the path to this leaf cannot by $H$-realizable because $L$ makes at most $d$ mistakes on $H$-realizable sample. ∎

Main result of this section is the the converse of this proposition is strongly false already for $d = 2$ (although, as we will see later, it is true fof $d = 1$).

**Theorem 7** *There exists a class $H$ of effective Littlestone dimension 2 which, for all $d$, does not have a partial computable online learner with at most $d$ mistakes.*

**Proof** We treat the set of functions $f \colon \mathbb{N} \to \{0,1\}$ as the Cantor space $\{0,1\}^{\mathbb{N}}$ with topology where open sets are unions of sets called "cylinders", and each cylinder is given by some sample $S$ and consists of all functions $f \colon \mathbb{N} \to \{0,1\}$, consistent with $S$.

In our construction, $H$ will have ordinary Littlestone dimension at most 2 and it will be "effectively closed", and these two things will guarantee that $H$ has effective Littlestone dimension. Now, "effectively closed" means that $H$ will be given by an enumerable set of "local restrictions" (of the form, "at these (finitely many) positions, you cannot have this combination of values"), and it will

consist of all function that satisfy all these prohibitions. Equivalently, we are giving the complement of $H$ as an enumerable union of cylinders.

Let us show why these two properties (ordinary Littlestone dimension at most 2 and effective closeness) imply that $H$ has effective Littlestone dimension at most 2. We have to provide an algorithm that, given a depth-3 Littlestone tree, gives a leaf of it such that the sample on the path from the root to this leaf is not $H$-realizable. We know that such leaf $\ell$ exists because $H$ has ordinary Littlestone dimension at most 2. The sample $S_\ell$, corresponding to $\ell$, is not consistent with any function from $H$. That is, the cone, corresponding to functions, consistent with $S_\ell$, belongs entirely to the complement of $H$, that is, it is covered by the enumeration of open sets, defining the complement of $H$. By compactness of the Cantor space, this means that this cone is covered by finitely many open sets in this enumeration. Hence, if we start running this enumeration, after finitely many steps we will find out for some leaf $\ell$ that the cone, induced by $S_\ell$, is already completely covered, meaning that we can output $\ell$.

We now give a construction of $H$. We fix a computable enumeration $L_1, L_2, L_3, \ldots$ of all partial computable learners. First, for a fixed $L_i$, let us give a construction of an effectively closed class $\widehat{H}_i$ with at most 2 functions that fools $L_i$ in the following sense: either (a) there is a sequence of examples from a function in $\widehat{H}_i$ on which $L_i$ makes infinitely many mistakes; or (b) there is a sample, consistent with a function in $\widehat{H}_i$, on which $L_i$ does not halt.

We first give 1 to $L_i$ for prediction, on the empty sample. We then start listing restrictions of the form "$f(1) = f(k)$" for $k = 2, 3$, and so on (forbidding different values at 1 and $k$). If $L_i$ never halts, we will list all such restrictions, forcing that $\widehat{H}_i$ consists of two constant functions. Then $L_i$ is fooled by not halting on the empty sample (realizable by $\widehat{H}_i$).

Assume now that $L_i$ halts on 1, predicting some value $b_1 \in \{0, 1\}$. At this moment, we have listed restrictions "$f(1) = f(k)$" for $k$ up to some $k_1 \in \mathbb{N}$. We then set a restriction $f(1) = \neg b_1$, leaving in $H_i$ exactly functions that are equal to $\neg b_1$ on $1, \ldots, k_1$ (with no restrictions so far on the other numbers). This forces the first mistake of $L_i$.

Next, we give $k_1 + 1$ for prediction to $L_i$, after the sample $(1, \neg b_1)$. While it is working, we start listing restrictions $f(k_1 + 1) = f(k_1 + k)$ for $k = 2, 3$, and so on. Once again, if $L_i$ never stops processing $k_1 + 1$, all restrictions like that will be listing, leaving in $\widehat{H}_i$ two functions: equal to $\neg b_1$ on $1, \ldots, k_1$ and constant on $\{k_1 + 1, k_1 + 2, k_1 + 3, \ldots\}$. In this case $L_i$ is fooled by not halting on the sample $(1, \neg b_1)$, realizable by both of this function.

Next, if $L_i$ at some moment stops processing $k_1 + 1$, predicting $b_2$, we set a restriction $f(k_1 + 1) = \neg b_2$, leaving in $\widehat{H}_i$ functions that are equal to $\neg b_1$ on $1, \ldots, k_1$ and are equal to $\neg b_2$ on $k_1 + 1, \ldots, k_1 + k_2$, where $k_2$ is the number of restrictions we have listed while $L_i$ was working on $k_1 + 1$.

We continue in the similar manner now with $k_1 + k_2 + 1$, and so on. There will be two possibilities. Either in the end $H_i$ will consist of two functions, equal to some fixed values $\neg b_1, \neg b_2, \neg b_3, \ldots$ on some initial blocks of natural numbers, and constant on the rest of natural numbers. This happens if $L_i$ never stops processing some realizable sample. The other option is we will eventually split all natural numbers into finite blocks, where inside the $j$th block the value of functions $f \in \widehat{H}_i$ is fixed to $\neg b_j$, forcing $\widehat{H}_i$ to have exactly 1 function that causes infinitely many mistakes when first numbers from the blocks are given $L_i$ for prediction.

We now have to combine this construction into a single effectively closed class $H$ of Littlestone dimension at most 2 that fools every $L_i$. We partition natural numbers into infinitely many infinite disjoint blocks in some computable way, assigning each $L_i$ one of the blocks. We will have two kind

of restrictions. First, for every pair of numbers from different blocks, we will forbid both of them having label 1, forcing every function in $H$ to have value 1 in at most one of the blocks. Restrictions of the second type will involve only numbers from the same block. Namely, for every $i$ in parallel, we list restrictions that fool $L_i$ as in the definition of the class $\widehat{H}_i$, but using the set of numbers of the $i$th block instead of $\{1, 2, 3, \ldots, \}$. Additionally, we do it in a slightly different way that adds the all-0 function to the class $\widehat{H}_i$ if it was not there already (every prohibition will involve at least one value 1 so that the all-0 function satisfies all of them).

Namely, every restriction that we have for $L_i$, saying "you cannot have these values in these positions", is turned into infinitely many restrictions, where for every $x$ from the $i$-th block, we say "you cannot have such values in such positions and have 1 at position $x$ simultaneously". Any function, satisfying old restrictions, satisfies all these new restrictions because the restrictions are weakened. On the other hand, any function $f$ with at least one value 1, violating some old restriction, will violate a new restriction where as $x$ we take some number on which $f$ is equal to 1.

We need this modification because we want the following property for our construction: every combination of values on the $i$-th block, satisfying all restrictions with the numbers of the $i$-th block, is extendable with all 0s to other blocks, without violating restrictions of the first type and restrictions inside other blocks. This property holds because all these other restrictions involve at least one value 1 outside the $i$-th block.

As a result, the class $H$ we obtain is as follows: it consists of the all-0 function and, for every $i$, of the functions that have ones only in the block $i$, and the labels in this block are an exact copy of the labels of one of the functions from $\widehat{H}_i$.

We conclude that, firstly, $H$ fools every partial learner $L_i$, meaning that $L_i$ either doesn't halt on some $H$-realizable sample, or it makes infinitely many mistakes on some sequence of examples from a function in $H$. And secondly, one can represent

$$H = H_1 \cup H_2 \cup H_3 \ldots$$

where $H_i$ is the set of functions from $H$ that have only 0s outside the $i$-th block. And $H_i$ consists of functions whose projection to the $i$-th block belong to $\widehat{H}_i$ (and that are 0 outside the $i$-th block) and the all-zeros function. In particular, besides the all-zeros function, $H_i$ has at most 2 functions, for every $i$. This means that the Littlestone dimension of $H$ is at most 2. Indeed, there is an online (non-computable) learner for $H$ with 2 mistakes, implying by Proposition 2 that $\mathsf{Ldim}(H) \leq 2$. This learner first predicts 0 on every number. If it's wrong, it is because there is a positive label in some block. This leaves the algorithm with at most 2 possible functions left. The learner first predicts according to one of them, and, in case of the second mistake, according to the second one. ∎

## 5. Equivalence in the bounded regime

On the positive side, we consider a modification of online learning where the learner initially gets an upper bound N on the numbers it will receive for prediction. It can be arbitrarily large, but the bound on the number of mistakes d should not depend on N. We call it *online learning in the bounded regime*. We show that effective Littlestone dimension characterizes computable learnability in this setting. As a corollary, we get the separation between computable online learning in bounded and

unbounded regime. For some class, online learning with bounded number mistakes is possible when the learner gets an arbitrary bound on the numbers, but not possible without a bound.

To be precise, we say that *L online learns H in the bounded regime with at most d mistakes* if, for any $N \in \mathbb{N}$ and any $H$-realizable sample $S = (x_1, y_1), \ldots, (x_k, y_k)$ with the property that $x_1, \ldots, x_k \leq N$, there are at most $d$ of $i \in \{1, \ldots, k-1\}$ such that $L(N, (x_1, y_1), \ldots, (x_i, y_i), x_{i+1}) \neq y_{i+1}$.

**Proposition 8** *A hypothesis class $H$ has effective Littlestone dimension at most $d$ if and only if there is a computable learner $L$ which online learns $H$ in the bounded regime with at most $d$ mistakes.*

**Proof** Assume that $L$ is a computable learner which online learns $H$ in the bounded regime with at most $d$ mistakes and let $T$ be a Littlestone tree of depth $d + 1$, where we have to output a leaf with the non $H$-realizable sample. We take as $N$ the largest number, appearing in $T$, and run the same procedure as in the proof of Proposition 6, with $L$ having $N$ as the additional input.

Next, assume that $H$ has effective Littlestone dimension at most $d$. Hence, there is an algorithm $A$ that, given a $(d+1)$-depth Littlestone tree $T$, outputs a leaf whose sample is not $H$-realizable. We construct a learner $L$ that, given $N$, goes through all Littlestone trees of depth $d+1$ with node labels at most $N$, computes all samples that are indicated by $A$ in these trees, and finds the set $H_N$ of all functions on the first $N$ natural numbers that are inconsistent with all these samples. The set $H_N$ includes all functions that can be continued to a function in $H$. On the other hand, the Littlestone dimension $H_N$ is at most $d$ as "witnessed" by $A$. The class $H_N$ is over a finite domain, and we have a complete description of it, so we find an online learner with at most $d$ mistakes for it by the brute-force. ∎

One could wonder whether the similar equivalence could be proven for finite classes of functions but exchanging "computable" for some of form of time-bounded computability, such as 'polynomial-time computable'. We leave this as an interesting direction for further research.

## 6. Effective Littlestone dimension and computability

**Theorem 9** *Let $H$ be a hypothesis class with finite effective Littlestone dimension. Then all functions in $H$ are computable.*

**Proof** We establish the theorem by induction on $\mathsf{effLdim}(H)$. When $\mathsf{effLdim}(H) = 0$, we have an algorithm that, given $x \in \mathbb{N}$, outputs $b \in \{0, 1\}$ such that $f(x) \neq b$ for all $f \in H$. Then $\neg b$ will be the value of $f(x)$ for the unique function of $H$, giving us an algorithm to compute this function.

For the induction step, we need the following lemma, which is an analog of the Proposition 3 for effective Littlestone dimension.

**Lemma 10** *For any class $H$ of finite positive effective Littlestone dimension, and for any $x \in \mathbb{N}$, either $H_0^x$ or $H_1^x$ have smaller effective Littlestone dimension that $H$.*

**Proof** Let $d = \mathsf{effLdim}(H) > 0$. There exists an algorithm $A$ that, given a $(d+1)$-depth Littlestone tree, outputs a leaf in it such that the sample on the path to this leaf is not $H$-realizable.

We now describe two algorithms, $A_0$ and $A_1$, and show that either $A_0$ establishes that $H_0^x$ has effective Littlestone dimension at most $d - 1$, or $A_1$ establishes that $H_1^x$ has effective Littlestone dimension at most $d - 1$. Namely, both algorithm receive on input a $d$-depth Littlestone tree (here

we need a condition $d > 0$ so that the notion of "$d$-depth trees" makes sense). The algorithm $A_0$ is supposed to, in any such tree, to output a leaf which is not $H_0^x$-realizable (this is a shortening of "the sample on the path from the root to this leaf is not realizable by $H_1^x$"). Likewise, $A_1$ is supposed to, for any such tree, output a leaf which is not $H_x^1$-realizable. We will show that this is true for at least one of the algorithms.

The algorithm $A_0$ works as follows. It receives a Littlestone tree $T_0$ of depth $d$ where it should output a not $H_0^x$-realizable leaf. For that, it goes over all depth-$d$ Littlestone trees $T_1$, and for each of them, does the following. It constructs a tree $T = (x, T_0, T_1)$, where the root is labeled by $x$, the 0-subtree coincides with $T_0$, and the 1-substree coincides with $T_1$. The algorithm gives this $T$ to $A$. If $A$ outputs a leaf in the 0-subtree of $T$, that is, inside $T_0$, the algorithm $A_0$ outputs this leaf as its answer and halts. Otherwise, $A_0$ proceeds to the next $T_1$.

Why is this algorithm correct? Samples that in $T$ are written on the paths to the leafs in the 0-subtree include the pair $(x, 0)$, written on the edge from $x$ (the first edge of this paths). When $A$ outputs a leaf $\ell$ in the 0-subtree, this means that the corresponding sample without $(x, 0)$ cannot be realized by $H_0^x$, otherwise adding a pair $(x, 0)$ we get an $H$-realizable sample. But when we remove $(x, 0)$ from the sample, we get exactly the sample, written on the path to $\ell$ from the root of $T_0$.

The problem with $A_0$ is that it might not halt on some $T_0$, when $A$ outputs a leaf in the 1-subtree for all $T_1$. We now define a similar algorithm $A_1$, receiving a depth-$d$ Littlestone tree $T_1$ (where it should indicate a not $H_1^x$-realizable leaf), and runs $A$ on all trees of the form $(x, T_0, T_1)$, waiting until $A$ indicates a branch in the 1-tree. By the same argument, this algorithm is correct.

The only case when both algorithms fail is when there exist $T_0', T_1'$ such that $A$ goes to the 1-subtree in all trees of the form $(x, T_0', T_1)$, and goes to the 0-subtree in all trees of the form $(x, T_0, T_1')$. However, this means that $A$ does not output anything in the tree $(x, T_0', T_1')$, a contradiction. ∎

Let us now finish the induction step. Let's say we have a class $H$ of effective Littlestone dimension $d > 0$, and for all smaller effective Littlestone dimensions, the theorem is already proved.

We take the algorithm $A$ that, given a $(d+1)$-depth Littlestone tree $A$, outputs a not $H$-realizable leaf. We first extend $H$ with all functions that "agree" with $A$. Namely, a function $f$ agrees with $A$ if in all trees $A$ outputs a leaf whose sample is inconsistent with $f$. By definition, all functions in $H$ agree with $A$, but potentially there are other function, which we all add to $H$. The resulting $H$ still has effective Littlestone dimension $d$ because of the same algorithm $A$. In other words, we consider the maximal class for which the algorithm $A$ works as a "witness" of effective Littlestone dimension $d$. The resulting superclass is effectively closed, because its complement is a union of cylinders and these cylinders can be effectively enumerated. Each of these cylinders corresponds to a forbidden path given by $A$ on some Littlestone tree $T$ and such trees, as finite objects, can be effectively enumerated.

For any $x \in \mathbb{N}$, one of the two restrictions $H_0^x$ and $H_1^x$ have effective Littlestone dimension smaller than $d$. In particular, by induction hypothesis, all functions in this restriction are computable. Thus, there is potentially just one function $f \in H$ missed in our considerations. Namely, this is a function $f$ such that for every $x \in \{0, 1\}$ we have $\mathsf{effLdim}(H_{f(x)}^x) = d$. It might be that such $f$ does not exist (if $\mathsf{effLdim}(H_0^x) < d$ and $\mathsf{effLdim}(H_1^x) < d$ for some $x$) or that $f \notin H$, but these are easy cases as in them there is nothing to prove. We therefore assume that such $f$ exists and belongs to $H$.

First, consider the case when there exists $n \in \mathbb{N}$ for which there is no $g \in H \setminus \{f\}$ such that $f(1) = g(1), \ldots, f(n) = g(n)$. In other words, there exists some finite sample from $f$ such that no other function from $H$ is consistent with this sample. Then $f$ can be computed as follows. First, into the program, we hardwire the values $f(1), \ldots, f(n)$. Then we give a procedure that computes $f(x)$ on a given $x > n$. Due to our condition, one of the two samples:

$$(1, f(1)) \ldots (n, f(n))(x, 0) \qquad (1, f(1)) \ldots (n, f(n))(x, 1)$$

is not realizable by $H$. And we can find out which one in finite time because $H$ is effectively closed. Indeed, one of the cylinders, corresponding to these two samples, belongs entirely to the complement of $H$, meaning that it is covered by a finite number of cylinders in the enumeration of this complement, by compactness of the Cantor space.

Assume now that every finite sample from $f$ is consistent with some function $g \in H$, different from $f$ (and thus, by the induction hypothesis, computable). We will use an online learning algorithm with "consistent oracle" (Kozachinskiy and Steifer, 2024; Assos et al., 2023). A consistent oracle for a class $H$ is a mapping that, given an $H$-realizable sample $S$, gives an function $f_S \in H$, consistent with this sample (more precisely, it gives an oracle access to it, meaning that given $S$ and $x \in \mathbb{N}$, it allows to evaluate $f_S(x)$). Kozachinskiy and Steifer (2024) give an algorithm that for any class $H$ of Littlestone dimension $d$, given only access to a consistent oracle for $H$, online learns it with at most $O(256^d)$ mistakes. This algorithm, to compute $L(S, x)$, the prediction on $x$ after the sample $S$, uses consistent oracle only for $S$ and its subsamples, making sure that it never applied to a not $H$-realizable sample.

We get back to the class $H$ in question, and we take any consistent oracle $H$ that never uses function $f$. Such oracle exists because any $H$-realizable sample is either inconsistent with $S$, so we cannot use $f$ for it in any case, or it is consistent, but then there is another function in $H$ which is also consistent with (and this function is computable!).

We consider the online learner $L$ of Kozachinskiy and Steifer (2024), equipped with this consistent oracle. By Lemma 1, there exists a sample $S$, consistent with $f$, such that $L_S$ coincides with $f$, that is, $L(S, x) = f(x)$ for all $x \in \mathbb{N}$. We use this to give a program for $f$, namely, we show that $L_S$ is computable. In computing $L(S, x)$ we use consistent oracle only to $S$ and its subsamples, so we oracle access to finitely many functions from $H$. Since the consistent oracle, by constructions, uses only computable functions, their programs can be hardwired into a program for $L_S$, emulating the computation of $L(S, x)$. ∎

**Corollary 11** *Let $H$ be a class of effective Littlestone dimension 1. Then it has a total computable online learner with at most 1 mistake.*

**Proof** Assume first that $H$ is finite. By Theorem 9, all finitely many functions of $H$ are computable. In this case, we can realize the standard optimal algorithm of Littlestone (1988) by a total Turing machine. In case when $\mathsf{Ldim}(H) = 1$, it works like this: given $x \in \mathbb{N}$, it takes $b \in \{0, 1\}$ such that $\mathsf{Ldim}(H_b^x) = 0$ (existing by Proposition 3)and predicts $\neg b$ so that when it is wrong, we are in $H_b^x$ where there is exactly one function. To realize this algorithm by a total Turing machine, we need to be able to decide, whether a sample is realizable, and whether it is realizable by exactly one function from $H$. We can achieve this by evaluating all functions from $H$ on the numbers from the sample.

From now on we assume that $H$ is infinite, and we make it effectively closed by adding, if necessary, all functions that agree with the algorithm $A$, "witnessing" that the effective Littlestone dimension of $H$ is 1. First, there is no $x$ such that $\mathsf{Ldim}(H_0^x) = \mathsf{Ldim}(H_1^x) = 0$ because otherwise $H = H_0^x \cup H_1^x$ has size at most 2. Therefore, we can define a function $f\colon \mathbb{N} \to \{0, 1\}$ by setting $f(x)$ such that $\mathsf{Ldim}(H_{f(x)}^x) = 1$. We claim that this function belongs to $H$. Indeed, if not, since $H$ is closed, some finite sample

$$S = (x_1, f(x_1)) \ldots (x_k, f(x_k)),$$

consistent with $f$, is not extendable to a function from $H$. But then $H$ can be represented as a union $H_{\neg f(x_1)}^{x_1} \cup \ldots \cup H_{\neg f(x_k)}^{x_k}$, where every of these $k$ sets has Littlestone dimension 0, by construction of $f$, meaning that in $H$ there are at most $k$ function.

Therefore, $f \in H$ and hence is computable by Theorem 9. We give a total computable online learner for $H$ that works as follows. On samples, consistent with $f$, it predicts according to $f$. Now, assume that we are given a sample, inconsistent with $f$. Then, by construction of $f$, we are in a restriction of $H$ of Littlestone dimension 0, which, thus, has at most one function. This means that for any $x \in \mathbb{N}$, either $S(x, 0)$ or $S(x, 1)$ is not extendable to a function in $H$. Given that, we start enumerating cones whose union gives the complement of $H$ (using that we have extended $H$ to an effectively closed class) until we find that cones we got so far already cover all extension of $S(x, b)$, for some $b \in \{0, 1\}$. We predict then $\neg b$ on $x$. This learners can make at most 1 mistake on $H$ (the first moment it obtains an example, inconsistent with $f$). ∎

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
