# OpenReview forum: "Effective Littlestone dimension"
_algorithmiclearningtheory.org/ALT/2025/Conference — ALT 2025_

### Official Review · Reviewer_foY3 · 2024-10-25
**This is a good paper, I recommand accept.**

**Rating:** 7
**Confidence:** 4

**Review:**

This paper studies the online learning of a class $ \mathcal{H} \subset \\{0,1\\}^{\mathbb{N}} $ with computable predictors and investigates the intrinsic complexity measures that characterize this learning paradigm. The paper introduces a notion named the "Effective Littlestone Dimension," defined as the maximum number $ d $ for which there exists an algorithm $ w $ such that, for any binary tree of depth $ d+1 $ with $\mathbb{N}$-valued labels, the algorithm $ w $ outputs a path in the tree that is not consistent with $ \mathcal{H} $.

The paper demonstrates that if a class $ \mathcal{H} $ is computably online learnable, then the Effective Littlestone Dimension is bounded. However, there also exists a class with an Effective Littlestone Dimension of 2 that is not computably online learnable. The paper further investigates several special cases where a bounded Effective Littlestone Dimension implies computable online learnability, such as when the instances are bounded and for classes with an Effective Littlestone Dimension of 1.

I have verified almost all the proofs, and they appear to be correct. I find the paper interesting and believe it is suitable for publication at ALT.

**Strengths**

The main result, in my opinion, is the counterexample showing that the Effective Littlestone Dimension does not imply computable online learning, strengthening the result by Hasrati and Ben-David (2023), which holds only for the classical Littlestone dimension.

The proof is also very elegant, leveraging the key concept of "effective closedness" of a class $ \mathcal{H} $. This concept can be described as follows (paraphrased from the paper):

1. Let $ \mathcal{\Phi} = \\{\phi_1, \phi_2, \dots\\} $ be a computationally enumerable set  of computable functions $\phi_i: (\mathbb{N} \times \\{0,1\\})^* \rightarrow \\{0,1\\} $.
2. A class $ \mathcal{H} $ is said to be "effectively closed" with respect to $ \Phi $ if, for any sample $ S \in (\mathbb{N} \times \\{0,1\\})^* $ such that $ \phi_i(S) = 1 $ for all $ i \in \mathbb{N} $, then $ S $ is realizable by $ \mathcal{H} $.

This implies that, for any sample $ S $ that is *not* consistent with $ \mathcal{H} $, there exists an algorithm to certify it. Therefore, for any "effectively closed" class $ \mathcal{H} $ with bounded (classical) Littlestone dimension, one can also bound the Effective Littlestone Dimension by searching for paths that violate one of the $ \phi_i $s. The proof then proceeds to construct an "effectively closed" set that is not computably online learnable yet has a (classical) Littlestone dimension of 2.

I also find other parts, such as the result that a bounded Effective Littlestone Dimension implies the functions in the class must be computable, to be quite interesting.

**Weaknesses**

1. The proof is sometimes difficult to follow, such as Theorem 7. I suggest that the authors define "effective closedness" mathematically rather than describing it in words. It would also be beneficial to first state the construction, then provide the proof so that readers can better understand the meaning of "compactness" and "effective closedness."

2. I am not sure why the notion of Cantor set is even mentioned in the paper. It seems that you not only use compactness but an "effective" version of compactness (i.e., where the open covering should be computationally enumerable).

3. The references in the paper are very limited; the authors should at least cite literature from inductive inference, such as T. Zeugmann and S. Zilles, "Learning recursive functions: A survey." Even though the current paper does not consider learning in the limit, I believe many of the proof ideas share a similar spirit. Additionally, I am not sure why "PEC-learning" is mentioned, as it is never used anywhere in the paper.

4. There are several typographical errors; I note a few here:
    - Last paragraph of page 9: $ x \in \\{0,1\\} $ should be $ x \in \mathbb{N} $.
    - Paragraph following Proposition 6: "this section is the the converse".

**Paper Award:**

No

---

> ### Author Response · Authors · 2024-11-22
> **On effective closeness**
>
> We thank the reviewer for pointing out places that need clarification. Let us clarify them.
>
>
>
> We use not the Cantor set but the Cantor space, the topological space where open sets are unions of cylinders. We use a well-known fact that the Cantor space is compact. We call a class effectively open if it is a union of a recursively enumerable set of cylinders. We call a class effectively closed if the complement to it is effectively open.
>
> By a local restriction we formally mean a complement to a cylinder (with the intuition of ``forbidding’’ to have values as in the sample, inducing the cylinder). Giving a class by an enumerable set of “local restrictions” thus means giving a class as an intersection of an enumerable set of complements to cylinders. A class given in this way is effectively closed by definition.
>
>
>
> We do not use  effective compactness. Rather, we implicitly use the following property, which is a consequence of the compactness of the Cantor space (with compactness understood in the usual sense). For every effectively open set X, the set of cylinders C such that $C\subseteq X$, is enumerable. It is proved as follows: there exists a recursive enumeration $C_1, C_2, C_3, ...$ of cylinders  such that $X = \bigcup_{i =1}^\infty C_i$. We enumerate all cylinders $C$ such that $C\subseteq \bigcup_{i =1}^N C_i$ for some $N$. By compactness of the Cantor space, since cylinders are clopen, this will enumerate all cylinders $C\subseteq \bigcup_{i =1}^\infty C_i = X$.
>
>
> We use this property when showing that effective closeness + ordinary Ldim at most 2 => effective Ldim at most 2. Given a depth-3 tree T, we start enumerating cylinders that are subsets of the complement to H, until we get a cylinder, induced by some leaf of T.

---

### Official Review · Reviewer_q79b · 2024-11-07
**Review of “Effective Littlestone Dimension”**

**Rating:** 7
**Confidence:** 5

**Review:**

This paper investigates online mistake-bound learning under the constraint that the learning rule must be computable. This is a natural and basic question in learning theory.

The authors introduce a computable variant of the Littlestone dimension, adapting the concept of effective dimensions specifically for online learning settings. They demonstrate that finite effective Littlestone dimension is necessary, though not sufficient, for computable online learning, and identify specific cases where it characterizes the optimal mistake bound. Furthermore, they establish that finite effective Littlestone dimension guarantees that the class consists only of computable functions.

The paper is well-written, with clear explanations and proofs that appear correct.

However, I believe the basic setting of (computable) online learning and the definition of online learnability are not satisfactorily addressed. The authors define a class as online learnable if there exists an absolute finite bound on mistakes, regardless of input sequence length. From a learning theory perspective, it’s more natural to consider settings where the number of mistakes is sublinear in the sequence length, with the learner aware of the “horizon” or length of the sequence. Indeed, Littlestone’s classical result provides a dichotomy: either the adversary can force Omega(|S|) mistakes when the class is not learnable, or the learner can guarantee O(1) mistakes when it is. However, I don’t think the dichotomy in this theorem should dictate the definition of online learnability, which should instead account for sublinear mistake bounds.

Will relaxing the definition of online learnability in this way affect the impossibility result? Could it be that, with this definition, the effective Littlestone dimension fully characterizes online learnability?

Additionally, considering randomized learners and whether the adversary should be restricted to be computable might lead to a more complete theory of computable online learning in this context.

**Paper Award:**

No

---

> ### Author Response · Authors · 2024-11-22
> **On learning with sublinear number of mistakes**
>
> The reviewer suggested considering online learning with a sublinear number of mistakes, and asked if the computable version of this learning is equivalent to finite effective Littlestone dimension.  The answer to this question is no. In fact, computable online learning with a sublinear number of mistakes is equivalent to computable online learning with uniformly bounded number of mistakes. Indeed, assume $H$ admits a computable online  learner L_1 with a sublinear bound on the number of mistakes. Then there is some $n_0$ such that, on every $H$-realizable sample of length $n_0$, this learner makes less than $n_0$ mistakes. Consider a learner L_2, maintaining the following invariant: after i mistakes there is a subsample $S^\prime$ of the current sample $S$ such that the length of $S^\prime$ is $i$ and $L_1$ makes $i$ mistakes on $S^\prime$. This invariant ensures that L_2 makes less than $n_0$ mistakes on every $H$-realizable sample. To maintain the invariant, $L_2$ predicts according to the prediction of $L_1$ after $S^\prime$. After we get the $(i+1)$st mistake, a pair with this mistake is appended to $S^\prime$, preserving the invariant.

---

### Official Review · Reviewer_be9W · 2024-11-10
**Computable learners in online learning: computable online learning implies a finite 'Effective Littlestone dimension,' but not vice versa**

**Rating:** 7
**Confidence:** 3

**Review:**

This paper studies the model of computable learners in online learning (mistake-bound model).

Context: Recently, the model of computable learning was studied within the PAC framework, where the key change is that we require the learning algorithm (mapping a sample $S$ and input $x$ to a prediction) to be computable by a Turing machine. In this case, the VC dimension no longer characterizes learnability. Delle Rose et al. (COLT’23) recently showed that an "effective" (computable) VC dimension addresses this issue: if the VC dimension is $d$, then there must exist a Turing machine that, given $d+1$ examples, outputs a labeling that cannot be realized by the original concept class.

A natural extension is to define a "computable" Littlestone dimension for the online setting. Specifically, given a binary tree of depth $L+1$ (where $L$ is the Littlestone dimension), there exists a Turing machine that outputs a labeling that is impossible for the concept class.

Contributions:

- Showing that online computable learning implies a finite effective Littlestone dimension.

- Proving that the converse is false: there exists a class with Littlestone dimension 2 for which no online computable learner exists.

- Showing that if we relax the online game by allowing an arbitrarily large but finite range for the inputs, then both directions hold. This also holds if the effective Littlestone dimension is $1$.

This paper addresses an interesting question with nontrivial analysis (for example, in the proofs of Theorems 7 and 9). I would be happy to see it accepted, though I believe the following writing issues should be addressed.

Writing improvements:

- It seems unusual that the online learning setting is not formally defined, given that it is the primary model in this paper.

- Definition of Littlestone Tree: The definition provided differs from the "standard" definition (see "A Theory of Universal Learning," STOC'21). Typically, when we say there is a Littlestone tree for $H$ of depth $d$, we mean that every path from root to leaf is realizable by $H$. In this paper, however, this is not the case. The tree has a similar full binary structure but does not require that every path is realized by $H$, which may be confusing.
Is that correct?

- It would be helpful to include an intuition or overview for the proof of Theorem 7. Since the tools used in the proof are not easy to digest, the writing would benefit from a clearer explanation (especially as there is room for this within the first 12 pages).

- Typo in Lemma 1: "an d" should be "and."

**Paper Award:**

No

---

> ### Author Response · Authors · 2024-11-19
> **Re: Definition of Littlestone tree**
>
> Correct, we used the term Littlestone tree for such object (full binary tree labelled by natural numbers). Our definition follows that of e.g. Raman et al. 'Apple Tasting: Combinatorial Dimensions and Minimax Rates' (COLT 2024) and indeed, it diverges from the convention used in Bousquet et al paper.

---

> ### Author Response · Authors · 2024-11-24
> **About Theorem 7**
>
> We agree that Theorem 7 can be explained in a clearer way: we will try to add an overview of the construction used in the proof to improve the intuitive understanding of the technical tools we have used.

---

### Author Rebuttal · Authors · 2024-11-24

We thank the referees for their valuable feedback on our paper. We have replied to each reviewer individually. Thank you again for your work.

---

### Meta-Review · Area_Chair_jFai · 2024-12-06

**Recommendation:** Accept
**Confidence:** 4

**Metareview:**

This paper proposes a notion of "effective Littlestone dimension", a computable version of the usual Littlestone dimension, which governs the mistake bound in online learning. The effective Littlestone dimension of a class H is defined as the smallest d such that there is a Turing machine that, given any labelled tree, can find a leave that is not consistent with any function in H. The authors show that this is a necessary condition for computable online learning, but that it is not sufficient. For the latter, they give a class with (effective) Littlestone dimension of just 2 that does not admit computable online learner. They also show that, if the effective Littlestone dimension is 1, then there exists a computable online learner.

This paper proposes a natural dimension for computable online learning and provide several fundamental results, some of which are arguably surprising. Due to this, we recommend acceptance.

**Paper Award:**

No